# New Hybrid PVC/PVP Polymer Blend Modified with Er_2_O_3_ Nanoparticles for Optoelectronic Applications

**DOI:** 10.3390/polym15030684

**Published:** 2023-01-29

**Authors:** Alhulw H. Alshammari, Majed Alshammari, Mohammed Ibrahim, Khulaif Alshammari, Taha Abdel Mohaymen Taha

**Affiliations:** 1Physics Department, College of Science, Jouf University, Sakaka P.O. Box 2014, Saudi Arabia; 2Physics Department, College of Science and Arts, Jouf University, Al-Gurayyat P.O. Box 756, Saudi Arabia

**Keywords:** polymer blend, Er_2_O_3_ nanoparticles, activation energy, nonlinear refractive index

## Abstract

Polymer blend hybrid nanocomposites are of great importance for future optoelectronic applications. This paper presents the preparation of new polymer blend hybrid nanocomposites based on PVC/PVP modified with Er_2_O_3_ nanoparticles. A low-cost solution casting method has been used to prepare the polymer nanocomposites at 0.0, 0.1, 0.3 and 0.6 wt% of Er_2_O_3_. X-ray diffraction (XRD), Fourier transform infrared (FTIR), Raman spectroscopy, and environmental scanning electron microscopy (ESEM) measurements have all been used to examine the impact of a varying wt% of Er_2_O_3_ on the structural and optical characteristics of PVP/PVC polymer blends. The PVC/PVP polymer blend and Er_2_O_3_ nanoparticles showed a strong interaction, which was validated by XRD, FTIR, and Raman spectrum investigations. The SEM micrographs showed a remarkable complexation among the components of the polymer nanocomposites. The activation energies for thermal decomposition of PVC/PVP doped with different Er_2_O_3_ concentrations were less than that of the pure polymer film. The linear and nonlinear refractive indexes, dispersion energy, optical susceptibility and the energy gap values were found to be Er_2_O_3_ concentration-dependent. With an increase in Er_2_O_3_ concentration to 0.1 and 0.3 wt%, the dispersion energy and nonlinear refractive index improved, and thereafter decreased when the concentration was further increased to 0.6For the film doped with 0.1 wt% Er_2_O_3_, the optical band gap (E_opt_) of the composite film enhanced by about 13%. The optical absorption measurements revealed clear improvements with the addition of erbium oxide. Higher refractive index values of PVC/PVP/Er_2_O_3_ films qualify the polymer blend as a cladding for electro-optic modulators. Our results indicated that the PVC/PVP/Er_2_O_3_ polymer films could be suitable for optoelectronic space applications.

## 1. Introduction

The current advances in polymer science and technology have enabled the production of multiple types of polymer nanocomposites that are used in different industrial applications. Polymer blends have recently emerged as highly promising materials for many applications including solar cells, sensors, energy storage, as well as water desalination [1,2,3,4]. This is because the mixture of two or more polymer matrices often exhibits novel properties that cannot be achieved with their individual components. Moreover, successful dispersion of metal oxide nanoparticles into polymer blend matrices, as guest fillers, has become an interesting subject of research due to its potential to improve many physical and chemical properties which, in turn, expands the range of their potential applications [5,6,7]. 

Among the various types of polymers, PVC has a large variety of applications because of its excellent physical properties [8,9]. Recently, PVC polymer nanocomposites have been proposed for optoelectronic applications due to their superior optical characters such as transmittance, fluorescence and tunable refractive index [10]. However, the use of PVC is limited due to its moderate flexibility and lower thermal stability. PVP on the other hand has excellent properties such as transparency, charge storage capacity, stability, ease of processing and biocompatibility, but its commercial prospects are limited due to its brittleness. Earlier reports have demonstrated the miscibility of PVP with PVC through a decreased association of PVP chains [11]. Moreover, the dehydrochlorination temperature of PVC was found to decrease by the addition of PVP, thereby decreasing the defects in the PVC matrix [12]. Bhavsar, V. et al. [13] reported the enhanced dielectric properties, mechanical flexibility, and degradability of PVC/PVP blend. This study reported that PVP/PVC blend films with 50%/50% weight concentration have enhanced mechanical flexibility, improved degradability, and modulated dielectric constants (2–3). Moreover, the mixture of PVC and PVP exhibited a tunable optical band gap and the IR spectrum of this blend (50%;50%) demonstrated the complexation reaction between PVC and PVP. These findings suggest that the PVC/PVP blend could be an attractive host for the deployment of different nanoparticles. Although there are a considerable number of studies on metal oxide or sulfide nanoparticle-doped polymer blends, the reports on nanocomposites based on PVC/PVP blend matrices are very scarce. Nevertheless, in a recent study, we reported the enhanced thermal stability of PVC/PVP by the addition of zinc ferrite (ZnFe_2_O_4_) nanoparticles [14]. More recently, we demonstrated that the incorporation of SrTiO3 nanofillers in PVC/PVP blend polymers had a dramatic impact on the optical absorption, optical susceptibility, and linear and nonlinear refractive indexes [15].

Er_2_O_3_ nanoparticles have drawn remarkable interest in many studies due to their high dielectric constant (10–14) [16] and characteristic fluorescence emissions (650 nm for ^4^F_9/2_ to ^4^I_15/2_ and 580 nm for ^4^S_3/2_ to ^4^I_15/2_) [17]. These properties rendered erbium oxide a potential material for optoelectronic applications such as optical sensors and waveguide amplifiers [18]. Various studies reported the impact of using Er_2_O_3_ as a nanofiller on the properties of various polymer nanocomposites. For example, the addition of Er_2_O_3_ nanoparticles was shown to improve the electrical, thermal and optical properties of PMMA polymers [19]. Similarly, the electrical conductivity of CMC/PVA was observed to be enhanced after the addition of Er_2_O_3_ nanoparticles [20]. Additionally, the controlled attachment of Er_2_O_3_ nanoparticles has led to significant improvements in the thermal and dielectric properties of PVDF [21].

In the present work, we aim to investigate the feasibility of modifying the PVC/PVP mixture with Er_2_O_3_ nanoparticles for possible optoelectronic applications. The preparation protocol follows the method of solution casting that includes mixing the two dissolved polymers before the addition of dissolved Er_2_O_3_ nanoparticles at room temperature. To obtain detailed information on the structural properties of these nanocomposites, complementary characterization techniques will be employed. The thermal stability of the triple mixture films will be studied using TGA analysis, whereas the linear and nonlinear optical behavior with variable Er_2_O_3_ content will be also discussed.

## 2. Materials and Methods

AnalaR grade PVC (MERK, Germany) and PVP (LOBACEMIE, India) polymer powders were used for the synthesis of PVC/PVP polymer blends. The first step involved in the preparation of the polymer blend films was dissolving certain amounts of PVC (0.9 g) and PVP (0.1 g) polymers in tetrahydrofuran (THF) solvent provided from CARLO ERBA, Italy. This process was carried out for 60 min at room temperature on a magnetic stirrer. In a separate step, Er_2_O_3_ nanoparticles were dispersed in THF solvent in different proportions. Finally, the Er_2_O_3_ solution was mixed with the polymer mixture solution under magnetic stirring for 60 min. In polypropylene dishes, the final solution was poured before being allowed to dry naturally. 

The Shimadzu XRD 7000 (λ = 1.54056 Å) generated diffractograms of PVC/PVP/Er_2_O_3_ films. A Shimadzu FTIR–Tracer 100 spectrometer was used to measure the ATR spectra between 399 and 2000 cm^−1^. Raman data were recorded on Hound Unchained spectrometer at 785 nm and a 5 s exposure time. On a Thermo Fisher Scientific Quattro ESEM environmental scanning electron microscope, ESEM images were collected for blend films. TGA scans for the polymer films at a heating rate of 10 °C/min were recorded on a thermogravimetric analyzer (Shimadzu TGA-51 Japan) within the range 30–600 °C. On an Agilent Cary 60 UV-Vis spectrophotometer, optical absorption data were measured in the wavelength range of 190–1000 nm.

## 3. Results and Discussions

### 3.1. Strucutral Characterization

Figure 1 displays the X-ray diffraction patterns of the PVC/PVP polymer film and PVC/PVP/Er_2_O_3_ nanocomposites. The remarkable crystalline quality of Er_2_O_3_ nanoparticles was demonstrated by the strong XRD peaks. The (211), (222), (400), (440), and (622) reflections of cubic Er_2_O_3_ (JCPDS No. 35-0734) correspond to the peak positions at 2*θ* = 20.3°, 29.1°, 33.7°, 48.5° and 57.7°, respectively [21]. The average nanocrystalline size of Er_2_O_3_ was found to be 39 nm using the Scherer equation, according to the formula [22,23,24,25]:(1)$$D=\frac{0.9\lambda }{\beta cos\theta }$$
where *λ* is the X-ray wavelength and *β* is the full width at half maximum (FWHM).

The two diffuse peaks at 19.0° and 25.0° in the spectra indicate the semicrystalline character of the PVC/PVP blend. However, the XRD patterns of PVC/PVP/Er_2_O_3_ nanocomposites indicate that increasing the concentration of Er_2_O_3_ does not change the semicrystalline structure. This finding agrees with previous work that small concentrations of nanofillers were not able to modify the polymer network structure [21]. ESEM cross-sectional micrographs were taken to verify the distribution of Er_2_O_3_ nanoparticles within the polymer blend films. The images in Figure 2 show a good complexity of nano-Er_2_O_3_ with the PVC/PVP network. Moreover, the images in Figure 2e,f support the characterization of a homogeneous dispersion of Er_2_O_3_ nanoparticles inside the polymer blend composites. 

The FTIR spectra of the PVC/PVP/Er_2_O_3_ blends with different Er_2_O_3_ concentrations are shown in Figure 3. At 1650 cm^−1^, the hydroxyl group bending vibrations were identified for all polymer films [26]. The bands at 1322 and 1495 cm^−1^ belong to vibrations of C−O and C=C [27]. The bending vibrations of C−H plane verified at positions 684, 833 and 957 cm^−1^ [27,28]. 

The vibrations at 1462 and 1290 cm^−1^ are generated because of C−N stretching and bending vibrations, while the C−Cl bond is responsible for the bands at 605 and 633 cm^−1^ [27]. Er_2_O_3_ has a metal-oxygen bond (M−O) shown at 557 cm^−1^ [29].

The Raman spectra of the nanocomposites at various Er_2_O_3_ concentrations are shown in Figure 4. Stretching vibrations of the C−Cl bonds induced the strong peaks at 635 and 695 cm^−1^. Furthermore, the stretching vibration of C−H and CH_2_ bonds generated the sharp peak at 2916 cm^−1^ [30]. The intensity of the band at 635 cm^−1^ of the PVC/PVP blend is increased upon doping with only 0.1 wt% of Er_2_O_3_. The intercalation of PVC/PVP/Er_2_O_3_ was proven with this level of quenching, particularly in the C−Cl region where the asymmetric stretching vibration of Er_2_O_3_ is most noticeable.

### 3.2. Thermal Stability Measurements

In order to examine the thermal decomposition of PVC/PVP/Er_2_O_3_ films, TGA and DTG tests were performed at 0.0, 0.1, 0.3 and 0.6 wt% of Er_2_O_3_. The TGA plots have three distinct stages of thermal decomposition, as shown in Figure 5a. The first degradation stage of the undoped film occurs in the temperature range of 300 to 502 K, while for the films doped with 0.1, 0.3 and 0.6 wt% of Er_2_O_3_, it appears in the ranges of 300 to 398 K, 300 to 503 K and 300 to 408 K, respectively. The second stage (main stage) appears for the pure PVC/PVP film in the range of 389 to 459 K and for the nanocomposite films in the ranges of 480 to 555 K, 485 to 551 K and 485 to 559 K, respectively. The third degradation stage occurs in the range of 746 to 786 K for the undoped films, while it appears for doped films in the ranges of 730 to 756 K, 733 to 766 K and 638 to 669 K, respectively. This is typically associated with the PVC/PVP polymer blend backbone. Figure 5b represents the derivative thermogravimetric (DTG) curve where the highest decomposition temperature, Tp, is shifted toward higher values.

For all samples, the increase of Er_2_O_3_ concentration improved the thermal stability of PVC/PVP polymers films. We also investigated the thermal kinetics of these samples in the main stage using the Coats–Redfern method via the formula [31,32]:(2)$$Ln\left[\frac{-Ln\left(1-\alpha \right)}{T^{2}}\right]=-\frac{E_{a}}{RT}-ln\frac{AR}{\varphi E_{a}}$$
where α =$\;\left[\frac{m_{i}-m_{t}}{m_{i}-m_{f}}\right]$ indicates the decomposition degree, *E_a_* refers to the activation energy of decomposition stage, *R* represents the gas constant, the reaction rate coefficient represented by *A* and *φ* denotes the heating rate. Figure 5c shows Coats–Redfern graphs of $Ln\left[\frac{-Ln\left(1-\alpha \right)}{T^{2}}\right]$ versus $\frac{1000}{T}$ for the nanocomposites films. The *E_a_* values can be easily extracted from the slope of the straight line as *E_a_* =2.303 *R*× slope, while the reaction rate coefficient A can be evaluated from the y-axis intercept.

Table 1 shows the thermal calculations for PVC/PVP with various concentrations of Er_2_O_3_ (0.0, 0.1, 0.3 and 0.6 wt%). It is shown that the activation energies values of PVC/PVP doped samples are less than that of the undoped film [33]. The reduced activation energy is attributed to the movement of pure polymer segments [34].

The thermodynamic parameters of entropy, ∆*S*°, enthalpy, ∆*H*, and Gibbs free energy, ∆*G*°, have been evaluated in the main degradation stage using the following equations [21]:(3)$$\Delta S{}^{\circ}=R\left[ln\left(\frac{hA}{k_{\beta }T_{p}}\right)-1\right]$$
(4)$$\Delta H=E_{a}-RT_{p}$$
(5)$$\Delta G{}^{\circ}=\Delta H-\Delta S{}^{\circ}T_{p}$$
where *h* and *k_β_* represent Plank’s and Boltzmann constants. Table 1 shows negative values of entropy (∆*S*°) for all samples, which means that the active state becomes less disordered in the studied polymer blend. The enthalpy (∆*H*) calculations show positive values, presumably due to the change of translational, vibrational or rotation states. Moreover, the Gibbs free energy values are also positive because the residue’s free energy and thermal decomposition is larger than the pure blend [35].

### 3.3. Optical Properties

Figure 6a shows the optical absorbance and transmittance of pure and Er_2_O_3_-doped PVC/PVP. The presence of an absorption band at 280 nm for the pure PVC/PVP blend film is attributed to π→π* interband electronic transitions [36,37]. The absorption edge is blue shifted towards lower wavelengths for the doped films due to an increase in the optical band gap that may be caused by the formation of cation and anion intermolecular bonds. It can therefore be concluded that increasing the concentration of Er_2_O_3_ nanoparticles is responsible for the band defect formation in PVC/PVP blend films [38].

Figure 6b shows the optical transmittance for pure and doped PVC/PVP blend films. The image shows a clear drop in the transmittance for films doped with 0.1 and 0.3 wt% Er_2_O_3_. This decrease in the optical transmittance is expected, since the dispersion of Er_2_O_3_ forms dense nanoparticles within the blend matrix which increases photon scattering [39]. However, the optical transmittance spectrum does not follow the same trend for the sample doped with 0.6 wt% Er_2_O_3_. This is due to defect formations in the PVC/PVP polymer chains caused by high Er_2_O_3_ concentration after filling the polymer clusters. 

The following Tauc’s formula (Equation (6)) was used to determine the values of the optical band gap (*E_opt_*) [40,41]:(6)$$\alpha h\upsilon =k{\left(hv-E_{g}\right)}^{x}$$
where α and *hυ* are the absorption coefficient and the incident photon energy, respectively, and *k* is a constant. The direct and indirect allowed transitions have the values of *x* = 0.5 and 2.0, respectively. The direct (*E_dir_*) and indirect (*E_ind_*) optical energy gaps are obtained from the intercept of straight lines at zero absorption as shown in Figure 7a,b. Doping the PVC/PVP film with only 0.1 wt% Er_2_O_3_ causes a significant increase in *E_dir_* and *E_ind_* values by 12.5 and 9.0 %, respectively, compared to those of the pure film. Further increasing the concentration to 0.3 wt% causes minor changes in these energy values, as shown in Table 2. In addition, the optical band gap (*E_opt_*) is drastically increased by ~13% for the doped film with 0.1 wt% Er_2_O_3_. The improvement of charge transfer at the polymer/filler interface increase the optical energy gap [42]. For the sample doped with the higher concentration of 0.6 wt% Er_2_O_3_, the values of *E_dir_*, *E_ind_* and *E_opt_* are slightly decreased, which indicate a narrower band gap at this concentration.

The optical reflectance measurements in Figure 8a show that the reflectance increases with the increase in Er_2_O_3_ content for the film doped with 0.1 and 0.3 wt% Er_2_O_3_. However, when the Er_2_O_3_ content increases to 0.6 wt%, the optical reflectance decreases compared to the 0.3 wt% Er_2_O_3_ sample. This behavior can be explained by defect formation due to the presence of high molecular weight Er_2_O_3_ in the polymer blend films. The optical reflectance values extracted from the plot shown in Figure 7a support the calculation of refractive index (n) as follows [43]:(7)$$n=\left(\frac{1+R}{1-R}\right)+\sqrt{\;\;\frac{4R}{{\left(1-R\right)}^{2}}-k^{2}\;}$$
where *R* denotes the optical reflectance and *k* denotes the extinction coefficient (*k* = αλ/4π). Figure 7b illustrates that the refractive index increases with the addition of 0.1 and 0.3 wt% Er_2_O_3_, which could be due to large clusters formed via Er_2_O_3_ condensation [44]. However, the *n* value decreases for the sample doped with 0.6 wt% Er_2_O_3_.

Nevertheless, the refractive index values determined in this work were found to be higher than those obtained in a previous study [44]. Since they are all above 1.65, they are very attractive for optoelectronic applications for example LEDs, emissive displays [45] photonic crystals [46,47] and solar cells [48].

The following formula is used to express the refractive index dispersion of PVC/PVP/Er_2_O_3_ films [49]:(8)$${\left(n^{2}-1\right)}^{-1}=\frac{E_{0}}{E_{d}}-\frac{1}{E_{0}E_{d}}\;{\left(h\upsilon \right)}^{2}$$
where *E*_0_ is the single-oscillator energy and *E_d_* is the dispersion energy [50]. *E*_0_ and *E_d_* can be calculated from Figure 8a ((*n*^2^−1) vs. ${\left(h\upsilon \right)}^{2})$ by identifying the slope and interception value with the y-axis. The *E_0_* and *E_d_* values were found to be Er_2_O_3_ concentration dependent, as given in Table 2. The *E*_0_ value varies in the range from 4.23 eV to 4.75 eV. It increases with increasing Er_2_O_3_ content to 0.1 and 0.3 wt% and then decreases for the film doped with 0.6% Er_2_O_3_ due to the electron transition probability within allowed electronic bands [51]. The *E_d_* value also varies in the range of 10 eV to 27.9 eV, depending on the concentration of Er_2_O_3_. Its variation trend is compatible with the values of *E_0_* and the optical band gap. The *E_d_* value increases rapidly for the PVC/PVP polymer films doped with 0.1 and 0.3 wt% Er_2_O_3_, which reflects the enhancement in optical transition strength between the polymer blend and Er_2_O_3_. This, on the other hand, indicates that a charge transfer between the PVC/PVP blend and Er_2_O_3_ took place [39]. The reduction in *E*_d_ value as the concentration increases to 0.6 wt% may be explained by the formation of defects in PVC/PVP chains. The dependency of *E_0_* and *E_d_* values on the doping concentration agrees with a previous study on PVC/SiO_2_ [52]. In our study, almost all optical parameters depend on the Er_2_O_3_ concentration, which can be considered as a key control for the energy of excitation in the PVC/PVP matrix composites. The static refractive index (*n*_0_) at (*hv*→0) can be evaluated by Equation (9) [53]:(9)$$n_{0}{}^{2}=\left(1+\frac{E_{d}}{E_{0}}\right)$$

The evaluation of *n*_0_ values for pure and doped PVC/PVP films with different Er_2_O_3_ concentrations is given in Table 2. Following all the optical characteristics presented in this work, the *n*_0_ values increase from 2.16 to 2.62 for PVP/PVC films doped with the two lowest concentrations (0.1 and 0.3 wt%) while it decreases for the sample doped with 0.6 wt%. This result could be very interesting from an industrial point of view, since higher n0 values are required for many applications.

Figure 8b shows the plot of optical dielectric loss, ε_2_, as function of photon energy, hυ, where ε_2_ is expressed as 2nk [54]. The energy gap values were determined by extrapolating the linear part of the curve to the intercept with the photon energy (*hυ*) axis in the plot of ε_2_ as a function of *hυ*. The value of the *hυ* axis intercept represents the real band gap energy [55], which is compatible with the values of E_dir_ for all content of Er_2_O_3_.

The oscillator strength (*f*) is calculated based on *E_d_* and *E*_0_ (see Equation (10)) [46], and the values associated with Er_2_O_3_ content are recorded in Table 2.
(10)$$f=E_{d}E_{0}$$

As seen in Table 3, the values of oscillator strength increase with an increase in the Er_2_O_3_ concentration. Furthermore, the optical moments ($M_{-1}\;and\;M_{-3}$) of PVC/PVP/Er_2_O_3_ polymer films can be computed from Equation (11) [56]:(11)$$E_{0}^{2}=\frac{M_{-1}}{M_{-3}}\;and\;\;E_{d}^{2}=\frac{M^{3}{}_{-1}}{M_{-3}}$$

Table 3 contains the values of M_−1_ and M_−3_, which improved as the content of Er_2_O_3_ varies from 0.0 to 0.6 wt%. Therefore, the strength of optical transitions is strengthened by this increase in optical spectrum moments. Moreover, Equation (12) was used to calculate the linear optical susceptibility ($\chi^{\left(1\right)}$) and the third order nonlinear optical susceptibility ($x^{\left(3\right)}$) of PVC/PVP/Er_2_O_3_ polymer blends [57]:(12)$$\chi^{\left(1\right)}=\frac{\frac{E_{d}}{E_{0}}}{4\pi }\;,\;\;\;x^{\left(3\right)}=6.82\times 10^{-15}{\left(\frac{E_{d}}{E_{0}}\right)}^{4}$$

The values of $\chi^{\left(1\right)}$ and $\chi^{\left(3\right)}$ recorded in Table 3 were enhanced upon the increase in Er_2_O_3_ content. In addition, the nonlinear refractive index (*n_2_*) was estimated according to Equation (13) [58]:(13)$$n_{2}=\frac{12\pi x^{\left(3\right)}}{n_{0}}$$

Table 3 shows the variation of nonlinear refractive index with the content of Er_2_O_3_ content for the PVC/PVP polymer matrix. It is widely accepted that the increase in the nonlinear refractive index is due to increased interactions between the photons and metal oxide nanoparticles embedded in the blend matrix.

## 4. Conclusions

We have successfully prepared PVC/PVP/Er_2_O_3_ polymer nanocomposites for the first time using a low-cost solution casting method. Characterization techniques such as XRD, FTIR, Raman and FESEM confirmed a strong complexation between the polymer blend and the dispersed Er_2_O_3_ nanofillers. The TGA measurements carried out in this work showed that the addition of Er_2_O_3_ leads to changes in *E_a_*, Δ*G*, ΔH and Δ*S* thermal kinetics of the polymer films. The addition of 0.1 wt% Er_2_O_3_ into the pure blend film causes a significant increase of *E_dir_* and *E_ind_* by 12.5 and 9.0 %, respectively. The values of dispersion energy and nonlinear refractive index increase as Er_2_O_3_ concentration increases to 0.1 and 0.3 wt%, thereafter decreasing when the content further increases to 0.6 wt%. We found that increasing the Er_2_O_3_ concentration to 0.1 and 0.3 wt% enhances the strength of optical transition between the polymer matrix and nanofillers. Our analysis concluded that the potential benefits of PVC/PVP/Er_2_O_3_ blend polymers can be realized with lower Er_2_O_3_ concentrations. To this end, PVC/PVP/Er_2_O_3_ polymer nanocomposites are expected to play a substantial role in optoelectronic space applications in the future.

## Figures and Tables

**Figure 1 polymers-15-00684-f001:**
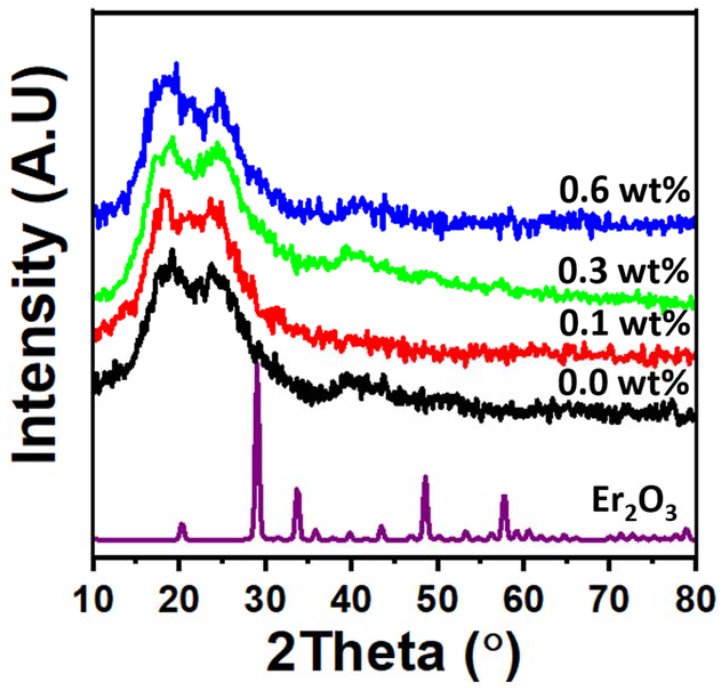
XRD diffraction peaks of PVC/PVP/Er_2_O_3_ blend nanocomposites.

**Figure 2 polymers-15-00684-f002:**
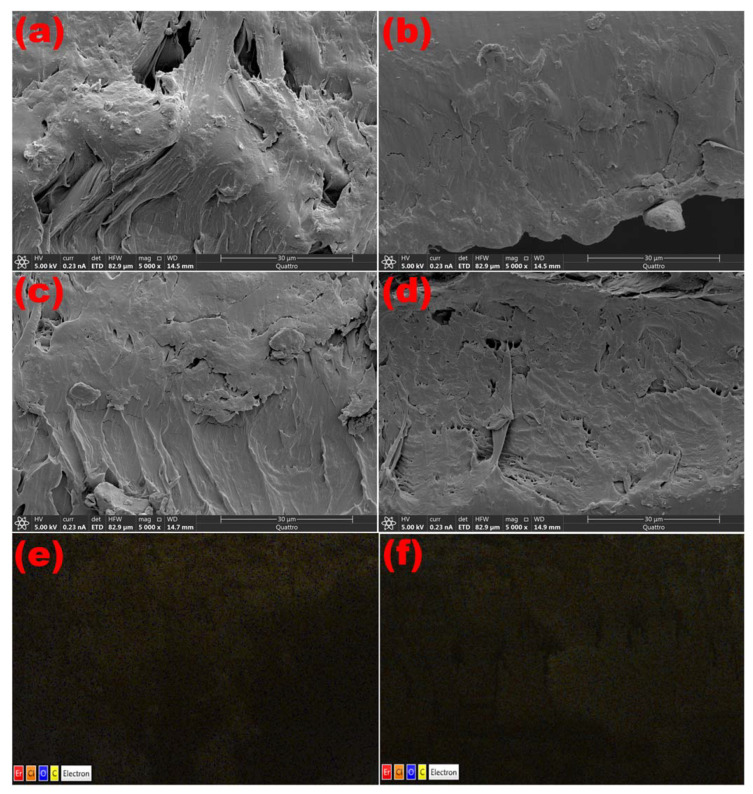
Cross-sectional ESEM scans of PVC/PVP/Er_2_O_3_ polymer films at (**a**) 0.0, (**b**) 0.1, (**c**) 0.3 and (**d**) 0.6 wt% and EDS mapping for (**e**) 0.3 and (**f**) 0.6 wt%.

**Figure 3 polymers-15-00684-f003:**
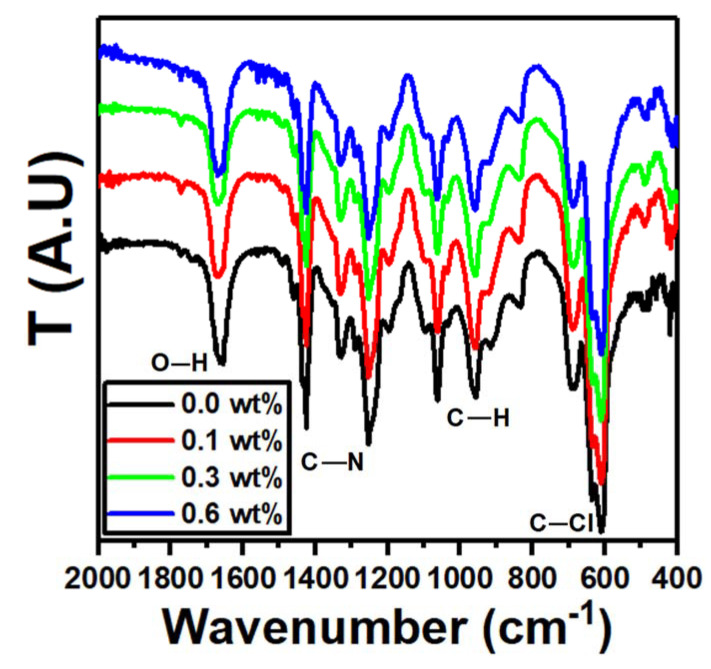
FTIR spectra of nanocomposites PVC/PVP/Er_2_O_3_.

**Figure 4 polymers-15-00684-f004:**
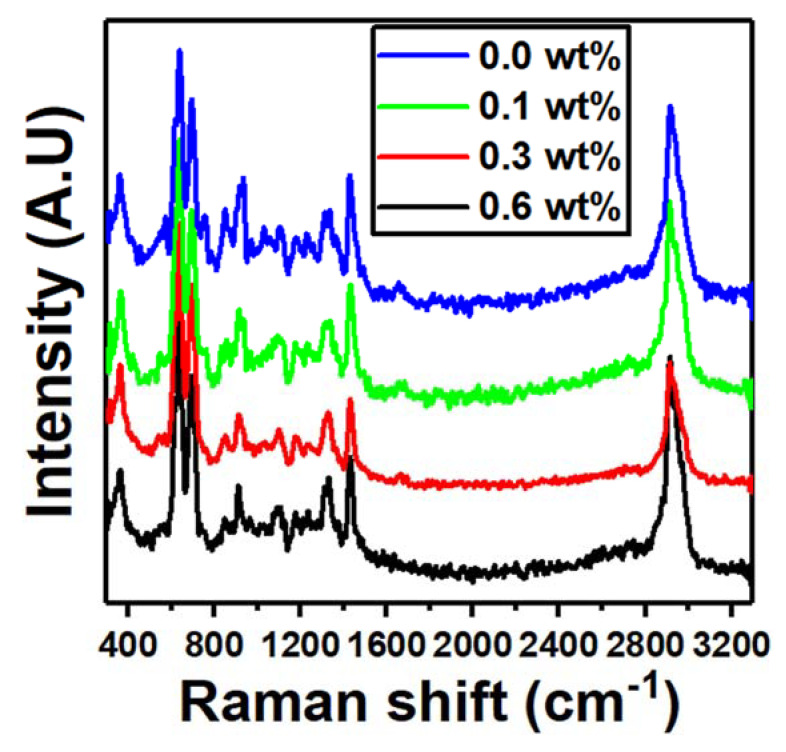
Raman spectra for blend nanocomposites PVC/PVP with different concentrations of Er_2_O_3_.

**Figure 5 polymers-15-00684-f005:**
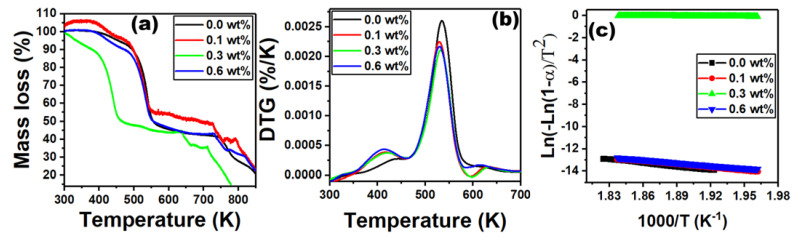
Representation of (**a**) TGA data plots, (**b**) DTG plots and (**c**) Coats–Redfern graphs for PVC/PVP doped with Er_2_O_3_ at different concentrations of 0.0, 0.1, 0.3 and 0.6 wt%.

**Figure 6 polymers-15-00684-f006:**
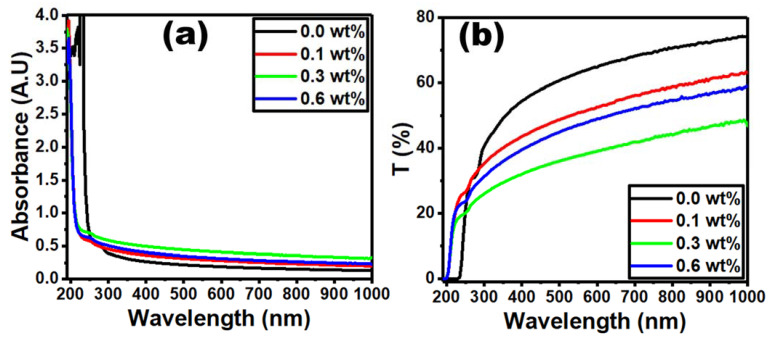
Plots of optical measurements for pure and doped PVC/PVP film with different Er_2_O_3_ concentrations: (**a**) absorbance and (**b**) transmittance against wavelength.

**Figure 7 polymers-15-00684-f007:**
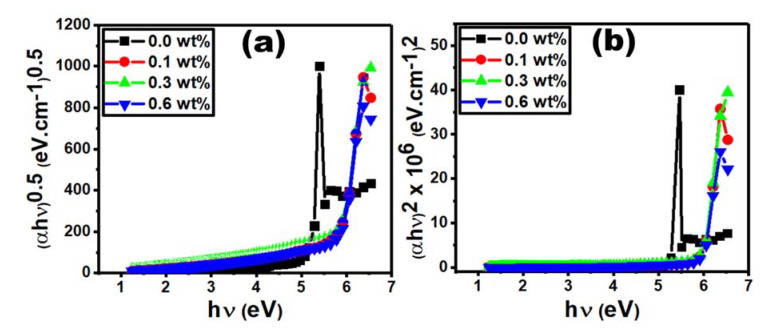
(**a**) (αhυ)^0.5^ vs. hυ and (**b**) (αhυ)^2^ vs. hυ for different Er_2_O_3_ concentrations.

**Figure 8 polymers-15-00684-f008:**
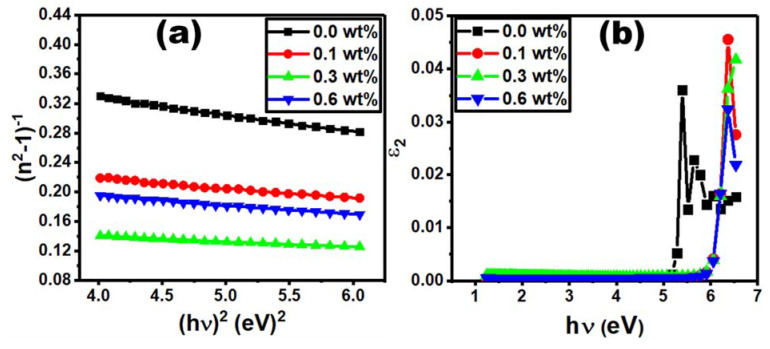
Plot of (**a**) reflectance and (**b**) refractive index for PVC/PVP films with different Er_2_O_3_ concentrations.

**Table 1 polymers-15-00684-t001:** The thermal parameters for PVC/PVP doped with different Er_2_O_3_ concentrations.

Er_2_O_3_ (wt%)	T_P_ (K)	E_a_ (kJ/mol)	A (S^−1^)	ΔS (J/mol.k)	ΔH (k/mol)	ΔG (kJ/mol)
0.0	523	212.75	2 × 10^8^	–98.91	208,396.15	260,172.8
0.1	527	174.55	5 × 10^6^	−130.28	170,171.48	238,829.1
0.3	533	10.63	16,395	−177.37	6200.39	100,740.5
0.6	529	163.79	2 × 10^6^	−138.51	159,392.49	232,666.8

**Table 2 polymers-15-00684-t002:** The optical parameters generated for the PVC/PVP/Er_2_O_3_ polymer films.

Er_2_O_3_ wt%	*E_dir_* (eV)	*E_ind_* (eV)	*E*_0_ (eV)	*E_d_* (eV)	*n* _0_	*f* (eV^2^)
0.0	5.28	5.25	4.23	10.0	1.83	42.32
0.1	6.0	5.78	4.47	16.35	2.16	73.05
0.3	5.99	5.82	4.75	27.89	2.62	132.45
0.6	5.98	5.69	4.41	18.0	2.25	79.03

**Table 3 polymers-15-00684-t003:** Values of optical moments, optical susceptibility, nonlinear refractive index and real optical energy gap for PVC/PVP/Er_2_O_3_ blend films.

Er_2_O_3_ wt%	M_−1_ (eV)	M_−3_ (eV)	χ ^(1)^ (esu)	χ ^(3)^ × 10^−13^ (esu)	n_2_ *×* 10^−12^ (esu)	*E_opt_* (eV)
0.0	2.36	0.13	0.19	2.13	4.38	5.27
0.1	3.66	0.18	0.29	12.24	21.37	6.05
0.3	5.87	0.26	0.47	81.08	116.60	6.03
0.6	4.08	0.21	0.32	18.96	31.71	6.02

## Data Availability

Data are available on request from the corresponding author.

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
