# Peer review of "New Hybrid PVC/PVP Polymer Blend Modified with Er_2_O_3_ Nanoparticles for Optoelectronic Applications"

_polymers, 2023, doi:10.3390/polym15030684_

Round 1

Reviewer 1 Report

Paper: New hybrid PVC/PVP polymer blend modified with Er2O3 nanoparticles for optoelectronic applications

This paper presents the preparation of a new polymer blend hybrid nanocomposites  based on PVC/PVP modified with Er2O3 nanoparticles. A low-cost solution casting method has been used to prepare the polymer nanocomposites at 0.0, 0.1, 0.3 and 0.6 wt% of Er2O.

The paper is well organized and interesting for readers. I recommend the publication of the paper with few minor changes.

1. In the Abstract part, the authors should provide the full names of the characterizations: FT-IR, XRD, etc.

2. Last line in abstract, our results indicated…instead of our results indicate

3. The purity of the reagents or polymers mentioned in the experimental part should be provided: PVP, PVC, etc.

4. I suppose that the thermodynamic parameters should be written as ΔS°, ΔG° instead of ΔS, ΔG.

5. Conclusion part, some verbs should be written in past form, confirm, etc.

Author Response

Response Letter to Reviewers’ Comments

Dear Reviewers,

We sincerely appreciate the valuable time the reviewers have spent reviewing our manuscript and providing insightful comments and suggestions to help further improve the quality of our work. Considering the reviewers’ evaluations, we have made a point-by-point response to the reviewers’ comments and revised our manuscript to improve the clarity of our work. We believe we have addressed all of the reviewers’ comments and now the paper is more rigorous in content and clearer in presentation. Our point-by-point responses to the reviewers’ comments are as follows.

Looking forward to hearing from you,

Sincerely,

Taha Hemaida

Response to Reviewer 1

  • In the Abstract part, the authors should provide the full names of the characterizations: FT-IR, XRD, etc.
  • We have provided the full names of the characterizations as in the revised manuscript.
  • Last line in abstract, our results indicated…instead of our results indicate
  • We have corrected the above word.
  • The purity of the reagents or polymers mentioned in the experimental part should be provided: PVP, PVC, etc.
  • We have provided the purity of polymers.
  • I suppose that the thermodynamic parameters should be written asΔS°,ΔG° instead of ΔS, ΔG.
  • We have corrected the above parameters.
  • Conclusion part, some verbs should be written in past form, confirm, etc.
  • We have corrected the conclusions.

All the corrections are highlighted in yellow.

Reviewer 2 Report

The manuscript has been written very well. However I have few comments as follow:

1) What is the ratio of PVC/PVP?

2) The quality of figures needs to be improved.

3) Please explain what is the activation energy described in equation 2. How to obtain the degree of decomposition?

Author Response

Response Letter to Reviewers’ Comments

Dear Reviewers,

We sincerely appreciate the valuable time the reviewers have spent reviewing our manuscript and providing insightful comments and suggestions to help further improve the quality of our work. Considering the reviewers’ evaluations, we have made a point-by-point response to the reviewers’ comments and revised our manuscript to improve the clarity of our work. We believe we have addressed all of the reviewers’ comments and now the paper is more rigorous in content and clearer in presentation. Our point-by-point responses to the reviewers’ comments are as follows.

Looking forward to hearing from you,

Sincerely,

Taha Hemaida

Response to Reviewer 2

  • What is the ratio of PVC/PVP?

  • The ratios of PVC and PVP were added to the revised manuscript.
  • The quality of figures needs to be improved.

  • The quality of figures has been improved.

  • Please explain what is the activation energy described in equation 2. How to obtain the degree of decomposition?
  • We have explained the activation energy and degree of decomposition in the revised manuscript.

All the corrections highlighted in yellow

Reviewer 3 Report

Alshammari et al. have presented the manuscript titled: New hybrid PVC/PVP polymer blend modified with Er2O3 nanoparticles for optoelectronic applications. Overall presentation of the article is good, but there require few modification before being publish, suggestions are as follow;

1.      Abstract of the article looks like explanatory, it is not defining the potential of the work very well, i.e., as authors have absorbance, band gap, reflectance etc, the best values of the results must be in abstract.

2.      In the abstract authors should describe among these Er2O3 additions, which concentration provided the best results.

3.      Important question is that why authors have empathized to select Er2O3 for this study, why not any else material?

4.      In introduction section, line 53, “its high dielectric constant and characteristic fluorescence emissions…”. Can authors describe these values with some reference?

5.      In the introduction portion, about the discussion of polymer part of material. literature survey is not strong. In my point of view if authors are writing, “Bhavsar, V. et, al reported an enhanced dielectric prop-45 erties, mechanical flexibility, and degradability of PVC/PVP blend…”, authors should also mention the values of results so that their results can be compared.

6.      Introduction is not strong, as it fails to provide the literature survey very well about the constituents, there must be values along with the results.

7.      In the material section, authors have not provided the detail about the materials they have used, from where they buy, manufacturer, purity etc.

8.      For XRD, SEM, FTIR, thermogravimetric analyzer and mass loss versus temperature, what parameters authors have used for measurements.

9.      In line 98, Authors have described,” increasing the concentration of Er2O3 does not change the semicrystalline structure….”Can authors describe the reason why there is not any change? Moreover, if authors use 0.8 or higher %, at what percentage the presence of Er2O3 is expected.

10.  “Two diffuse peaks at 19.0° and 25.0°…” Can authors index these positions and with which PDF card # it can be compared well?

11.  SEM image is not well defining the presence of Er2O3 nanoparticles in the material. I suggest the authors to provide the EDX or colored FESEM images to identify the presence of Er2O3 nanoparticles.

12.  Figure 3 will become more attractive if authors can add the C-O or C-N, C-N and C=C, and C-113 vibrations at the specific positions into the figure as well.

Author Response

Response Letter to Reviewers’ Comments

Dear Reviewers,

We sincerely appreciate the valuable time the reviewers have spent reviewing our manuscript and providing insightful comments and suggestions to help further improve the quality of our work. Considering the reviewers’ evaluations, we have made a point-by-point response to the reviewers’ comments and revised our manuscript to improve the clarity of our work. We believe we have addressed all of the reviewers’ comments and now the paper is more rigorous in content and clearer in presentation. Our point-by-point responses to the reviewers’ comments are as follows.

Looking forward to hearing from you,

Sincerely,

Taha Hemaida

Response to Reviewer 3

  • Abstract of the article looks like explanatory, it is not defining the potential of the work very well, i.e., as authors have absorbance, band gap, reflectance etc, the best values of the results must be in abstract.
  • We have added the best values to the abstract.
  • In the abstract authors should describe among these Er2O3 additions, which concentration provided the best results.
  • We have provided the best results.
  • Important question is that why authors have empathized to select Er2O3 for this study, why not any else material?
  • We have explained the importance of Er2O3 nanoparticles in the introduction.
  • In introduction section, line 53, “its high dielectric constant and characteristic fluorescence emissions…”. Can authors describe these values with some reference?
  • We have added the values as in the revised manuscript.
  • In the introduction portion, about the discussion of polymer part of material. literature survey is not strong. In my point of view if authors are writing, “Bhavsar, V. et, al reported an enhanced dielectric prop-45 erties, mechanical flexibility, and degradability of PVC/PVP blend…”, authors should also mention the values of results so that their results can be compared.
  • We have considered the above comment in the revised manuscript.
  • Introduction is not strong, as it fails to provide the literature survey very well about the constituents, there must be values along with the results.
  • We have considered the above comment in the revised manuscript.
  • In the material section, authors have not provided the detail about the materials they have used, from where they buy, manufacturer, purity etc.
  • Materials details were added to the revised manuscript.
  • For XRD, SEM, FTIR, thermogravimetric analyzer and mass loss versus temperature, what parameters authors have used for measurements.
  • All the parameters were added.
  • In line 98, Authors have described,” increasing the concentration of Er2O3 does not change the semicrystalline structure….”Can authors describe the reason why there is not any change? Moreover, if authors use 0.8 or higher %, at what percentage the presence of Er2O3 is expected.
  • The above comment was addressed in the revised manuscript.
  • “Two diffuse peaks at 19.0° and 25.0°…” Can authors index these positions and with which PDF card # it can be compared well?
  • The two diffuse peaks at 19.0° and 25.0° correspond to semicrystalline structure of PVC/PVP polymer blend.
  • SEM image is not well defining the presence of Er2O3 nanoparticles in the material. I suggest the authors to provide the EDX or colored FESEM images to identify the presence of Er2O3 nanoparticles.
  • The EDX data with mapping were added to the revised manuscript.
  • 12.  Figure 3 will become more attractive if authors can add the C-O or C-N, C-N and C=C, and C-113 vibrations at the specific positions into the figure as well.
  • The positions were added to fig. 3.

All the corrections highlighted in yellow